# Causal Discovery from Discrete Data using Hidden Compact Representation

**Ruichu Cai [1], Jie Qiao[1], Kun Zhang[2], Zhenjie Zhang[3], Zhifeng Hao[1, 4]**

[1] School of Computer Science, Guangdong University of Technology, China
[2] Department of philosophy, Carnegie Mellon University
[3] Singapore R&D, Yitu Technology Ltd.
[4] School of Mathematics and Big Data, Foshan University, China
cairuichu@gdut.edu.cn, qiaojie.chn@gmail.com, kunz1@andrew.cmu.edu,
zhenjie.zhang@yitu-inc.com, zfhao@gdut.edu.cn

## Abstract

Causal discovery from a set of observations is one of the fundamental problems across several disciplines. For continuous variables, recently a number of causal discovery methods have demonstrated their effectiveness in distinguishing the cause from effect by exploring certain properties of the conditional distribution, but causal discovery on categorical data still remains to be a challenging problem, because it is generally not easy to find a compact description of the causal mechanism for the true causal direction. In this paper we make an attempt to find a way to solve this problem by assuming a two-stage causal process: the first stage maps the cause to a hidden variable of a lower cardinality, and the second stage generates the effect from the hidden representation. In this way, the causal mechanism admits a simple yet compact representation. We show that under this model, the causal direction is identifiable under some weak conditions on the true causal mechanism. We also provide an effective solution to recover the above hidden compact representation within the likelihood framework. Empirical studies verify the effectiveness of the proposed approach on both synthetic and real-world data.

## 1 Introduction

Because randomized controlled experiments are usually infeasible and generally too expensive, observational data-based causal discovery, has been a focus of recent research in this area [Spirtes *et al.*, 2000; Pearl, 2009]. Various observational-based causal discovery methods have been proposed by exploring certain properties of the conditional distribution. For example, constraint-based methods exploit conditional independence relations between the variables in order to estimate the Markov equivalence class of the underlying causal graph [Spirtes *et al.*, 2000; Pearl and Verma, 1995]. On linear non-Gaussian acyclic data, the Linear, Non-Gaussian, Acyclic Model (LiNGAM) [Shimizu *et al.*, 2006, 2011] has been used to reconstruct the causal network by maximizing the independence among the noises. On nonlinear data, additive noise model [Hoyer *et al.*, 2009] and post-nonlinear model [Zhang and Chan, 2006; Zhang and Hyvärinen, 2009] can be used to distinguish the cause from effect by considering the independence between the noise and the cause. Recently, the likelihood embedded with various constraints and models among the variables are conducted for score-based methods [Cai *et al.*, 2018].

Though the additive noise model ($Y = g(X) + E, X \perp\!\!\!\perp E$) has been extended to handle discrete data [Peters *et al.*, 2010], causal discovery on categorical data still remains to be a challenging problem. Note that it is usually hard to justify the additive noise assumptions for discrete data, especially for

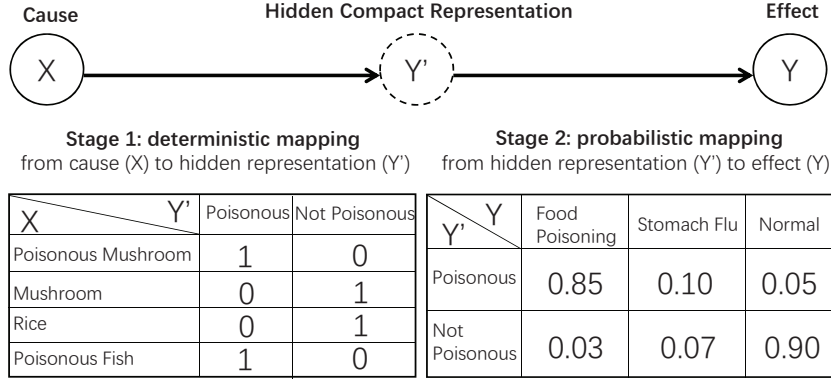

Figure 1: Food Poisoning: A Hidden Compact Representation Example in Real World.

categorical data. In fact, the additive noise model assumes that all categories of the variables are placed in the "right" order; furthermore, if $X \to Y$ holds according to the additive noise model, then for any observation $(x, y)$, there exists a function $g(X)$ such that the noise $E = y - g(x)$ is independent of $X$, i.e., it has the same distribution for different values of $x$. In other words, the conditional distribution $P(Y|X = x)$ always has the same shape for different values of $x$, after being properly shifted according to $g(x)$. But the values of a categorical variable are mutually exclusive categories or groups, without a meaningful order of magnitude. Thus, the additive noise model may not be a proper representation of the causal mechanism for categorical variables.

Therefore, a proper description of the causal mechanism for discrete data that helps in causal discovery remains under explored. In this work, we make an attempt to find a way to solve this problem by introducing a new assumption, called Hidden Compact Representation (HCR in short) as shown in the food poisoning example given in Figure 1: the first stage maps cause to a hidden variable of a lower cardinality, and the second stage generates the effect from the hidden representation. As shown in the example, in the first stage, the food $(X)$ with four different categories is mapped to the binary compact hidden representation $(Y')$ with the key information whether it is poisonous; in the second stage, the hidden representation (poisonous or not) determines whether the patient is diagnosed as having food poisoning $(Y)$. The hidden representation $(Y')$ provides a compact representation of the causes and captures key information of the causal mechanism, leaving out irrelevant information in the cause. This way, the causal mechanism admits a simple, compact representation.

Let us have a closer look at the hidden compact representation model and see whether it is possible to estimate it from data. First, these two stages are separated by the hidden representation, i.e., $X \perp\!\!\!\perp Y|Y'$ holds. As a result, we can use two conditional probabilities to express the whole causal mechanism from $X$ to $Y$, as shown in the tables in Figure 1. Second, as a compact representation of the cause, the first stage is deterministic, all stage transfer is done with probability 1. Third, as a causal mechanism, the second stage can be represented by a probabilistic mapping from the hidden variable $Y'$ to the effect $Y$. Based on the above observations, we provide a practical method to estimate the above HCR model under the likelihood framework. We also theoretically show that the model is identifiable under weak conditions on the causal mechanism.

Our main contributions include 1) proposing a two-stage compact representation of the causal mechanism in the discrete case, 2) developing a likelihood-based framework for estimating the HCR model, and 3) conducting a theoretical analysis of the identifiability of the underlying causal direction.

## 2 Hidden Compact Representation Model

Without loss of generality, let $X$ be the cause of $Y$ in a discrete cause-effect pair, i.e., $X \to Y$. Here, we use the hidden compact representation, $M : X \to Y' \to Y$, to model the causal mechanism behind the discrete data, with $Y'$ as a hidden compact representation of the cause $X$.

In the first stage $X \to Y'$, cause $X$ is mapped to a low-cardinality hidden variable $Y'$ deterministically. It can be expressed by using $Y' = f(X)$, where $f : \mathbb{Z} \to \mathbb{Z}$ is a noise-free arbitrary function. It

implies that the cause $X$ can be reduced to a hidden low-cardinality space $Y'$. This stage extracts the real, necessary causal factor behind the various cause states. As shown in Figure 1, there are four different values of $X$, and the hidden representation $Y'$ extracts the key information in this causal mechanism, i.e., whether the food is poisonous or not.

In the second stage, the effect $Y$ is generated from the hidden representation $Y'$ by the probabilistic mapping with conditional probability distribution $P(Y = y|Y' = f(x))$. For instance, as shown in Figure 1, the food poisoning may misdiagnose as stomach flu with probability 0.1, which is captured by the conditional distribution.

In this hidden compact representation model, the deterministic mapping stage and probabilistic mapping stage are naturally separable by the hidden representation $Y'$, i.e., $X \perp\!\!\!\perp Y|Y'$. Given a group of observations $\mathcal{D} = \{(x_i, y_i)\}_{i=1}^m$, the log-likelihood of the model $M : X \to Y' \to Y$ is estimated as follows.

$$
\begin{aligned}
&\mathcal{L}(M; \mathcal{D}) \\
&= \log \prod_{i=1}^m \sum_{y_i'} P(X = x_i, Y' = y_i', Y = y_i|M) \\
&= \log \prod_{i=1}^m \sum_{y_i'} P(X = x_i)P(Y' = y_i'|X = x_i)P(Y = y_i|Y' = y_i') \\
&= \log \prod_{i=1}^m P(X = x_i)P(Y = y_i|Y' = f(x_i))
\end{aligned}
\tag{1}
$$

Equation (1) decomposes the joint probability into three components according to $X \perp\!\!\!\perp Y|Y'$. The middle term of the second equation, $P(Y' = y_i'|X = x_i)$, denotes how the compact representation $Y'$ is generated from $X$. Since this process is deterministic, we have $P(Y' = y_i'|X = x_i) = 1$ if $y_i' = f(x_i)$ and $P(Y' = y_i'|X = x_i) = 0$ if $y_i' \neq f(x_i)$, where function $f$ denotes the true mapping.

Different from the previous likelihood framework, the likelihood given in equation (1) contains a hidden representation with an unknown cardinality. Thus, the Bayesian Information Criterion (BIC) [Schwarz and others, 1978] is introduced to control the complexity of the model, which provides a trade-off between the goodness of fit and model complexity. The BIC is given in equation (2), which is an approximation of the marginal likelihood of the hidden compact representation model $M$ based on the data $\mathcal{D}$:

$$
\mathcal{L}^*(M; \mathcal{D}) = \mathcal{L}(M; \mathcal{D}) - \frac{d}{2} \log(m)
\tag{2}
$$

where $d = (|X| - 1) + |Y'|(|Y| - 1)$ measures the effective number of parameters in the model. In detail, $(|X| - 1)$ and $|Y'|(|Y| - 1)$ are the numbers of parameters for $P(X)$, and the probabilistic mapping $P_{Y|Y'}$, respectively.

To recover the causal model, we regard the model with the highest $\mathcal{L}^*$ as the best one. The parameters in $M$ are decomposed into two parts, $\theta$ and $f$, where $\theta$ includes the parameters of $P(X)$ and $P(Y|Y')$. Maximization of above objective function, i.e., $\max \mathcal{L}^* = \sup_f \max_\theta \mathcal{L}^*$, involves two iterative steps. First, calculate the maximum likelihood estimator (MLE) $\hat{\theta} = \mathrm{argmax}_\theta \mathcal{L}(\theta; \mathcal{D})$ while fixing the function $f$. Second, fix the parameter values of $\theta$ and find a better model by choosing the best $f$ to achieve $\sup_f \mathcal{L}^*(f; \mathcal{D})$. Such an alternate maximization procedure eventually converges, as shown in [Bezdek and Hathaway, 2003].

In the first step, more specifically, while fixing the $f$, the maximization is equivalent to $\max_\theta \mathcal{L}^* = \max_\theta \mathcal{L}(\theta; \mathcal{D})$ and the MLE $\hat{\theta}$ of the likelihood $\mathcal{L}$ can be calculated directly as described in the following.

Let $\hat{\theta} = \{\hat{\mathbf{a}}, \hat{\mathbf{b}}\}$ where $\hat{a}_x = \hat{P}(X = x), \hat{b}_{y,y'} = \hat{P}(Y = y|Y' = y')$ denote the MLE of the distribution $P_X, P_{Y|Y'}$ respectively. The MLE of those parameters can be written as $\hat{a}_x = \frac{n_x}{\sum_x n_x}, \hat{b}_{y',y} = \frac{n_{y',y}}{\sum_y n_{y',y}}$, where $n_x = \sum_{i=1}^m I(x_i = x)$, and $n_{y',y} = \sum_{i=1}^m I(y_i' = y', y_i = y)$ are the frequencies of value $X = x$ and $Y' = y', Y = y$ in samples respectively. Such solutions can be derived by maximizing $\mathcal{L}$

with the constraints conditions $\sum_x a_x = 1$, and $\sum_y b_{y',y} = 1$, $\forall y'$. Following likelihood of the model in equation (1), the solution of $\max_\theta \mathcal{L}(\theta; \mathcal{D})$ is given in equation (3).

$$
\begin{aligned}
&\max_\theta \mathcal{L}(\theta; \mathcal{D}) \\
&= \log \prod_{i=1}^{m} P(X = x_i) P(Y = y_i | Y' = f(x_i)) \\
&= \log \prod_x \hat{a}_x^{n_x} \prod_{y'} \prod_y \hat{b}_{y',y}^{n_{y',y}} \\
&= \sum_x n_x \log(\frac{n_x}{\sum_x n_x}) + \sum_{y'} \sum_y n_{y',y} log(\frac{n_{y',y}}{\sum_y n_{y',y}})
\end{aligned}
\tag{3}
$$

In the second and third equalities, we collect each category and perform a MLE to estimate each parameter in the distribution. Consequently, the optimum solution $\hat{\theta}$ is given in the closed form, which will be used in the second step of the optimization procedure.

In the second step, we propose a greedy search algorithm to search the best $f$ with $\sup_f \mathcal{L}^*$. Firstly, $f(x)$ is initialized with the $y_0$, where $y_0$ is the mode of $\{y| < x, y > \in D\}$. Secondly, we perform greedy search on the $f(x)$ by enumerating all possible values for each $x$.

In summary, the optimization of $\max \mathcal{L}^* = \sup_f \max_\theta \mathcal{L}^*$ is given in the following algorithm.

---

**Algorithm 1** Optimization of $\max \mathcal{L}^* = \sup_f \max_\theta \mathcal{L}^*$

---

**Input:** Data $\mathcal{D}$
**Output:** $\mathcal{L}^*$
1: $f^{(0)}(x) \leftarrow \mathrm{argmax}_y \hat{P}(X = x, Y = y)$
2: **while** $\mathcal{L}^*$ no longer increases **do**
3:      $t = t + 1$
4:      **for** each pair $< x, y' >$ **do**
5:          $\mathcal{L}^* \leftarrow \max_\theta \mathcal{L}(f_{x \to y'}^{(t-1)}, \theta; \mathcal{D}) - \frac{d}{2} \log(m)$
6:          **if** $\hat{\mathcal{L}}^* < \mathcal{L}^*$ **then**
7:              $\langle \hat{x}, \hat{y}', \hat{\mathcal{L}}^* \rangle \leftarrow \langle x, y', \mathcal{L}^* \rangle$
8:          **end if**
9:      **end for**
10:     Set $f(\hat{x})$ to be $\hat{y}'$ and let $\mathcal{L}^* = \hat{\mathcal{L}}^*$
11: **end while**
12: **return** $\mathcal{L}^*$

---

where $f_{x \to y'}^{(t-1)}$ denotes that the change of the value $f^{(t-1)}(x)$ to value $y'$. In each iteration, we search the best gain pair $< \hat{x}, \hat{y}' >$ by traversing the value of $y'$ in $f_{x \to y'}^{(t-1)}$. In other words, change the value $f(\hat{x})$ to $\hat{y}'$ in order to achieve the highest score increase. Finally, set $f(\hat{x}) \leftarrow \hat{y}'$ and update the score until $\mathcal{L}^*$ no longer increases.

Based on the above proposed hidden compact representation and its BIC score, we can simply get the following practical method for causal inference.

1. Estimate the model $M : X \to Y' \to Y$, $\tilde{M} : Y \to X' \to X$ by maximizing $\mathcal{L}^*(M; \mathcal{D})$, $\mathcal{L}^*(\tilde{M}; \mathcal{D})$ respectively;

2. If $\mathcal{L}^*(M; \mathcal{D}) > \mathcal{L}^*(\tilde{M}; \mathcal{D})$, infer "$X \to Y$",
   If $\mathcal{L}^*(M; \mathcal{D}) < \mathcal{L}^*(\tilde{M}; \mathcal{D})$, infer "$Y \to X$",
   If $\mathcal{L}^*(M; \mathcal{D}) = \mathcal{L}^*(\tilde{M}; \mathcal{D})$, infer "non-identifiable".

The asymptotic correctness of this practical methods is implied by the identifiability of the model, which is theoretically analyzed in the following section.

# 3 Identifiability

We shall show that under the hidden compact representation model, the causal direction is asymptotically identifiable in the general case (under some technical conditions).

We first show the following property for the reverse direction under certain conditions on the conditional distribution $P(Y|X)$.

**Theorem 1.** *Assume that for the correct causal direction, the conditional distribution $P(Y|X)$ is random in the sense that*

> *A1. there does not exist values $y_1 \neq y_2$ such that $P(Y = y_1 \mid X)$ equals $P(Y = y_2 \mid X)$ times a constant for all possible $X$ values. (Note that both $P(Y = y_1 \mid X)$ and $P(Y = y_2 \mid X)$ are functions of $X$.)*

*Then asymptotically, in the reverse direction there does not exist $X' = \hat{f}(Y)$ with $|X'| < |Y|$ such that $P(X|Y) = P(X|X')$ for all possible $X$ and $Y$ values, i.e., the reverse direction does not admit a low-cardinality hidden representation $\hat{f}(Y)$.*

*Proof.* We have $P(X, Y) = P(X)P(Y|X)$ for the correct direction. Assume that there exists such a $X' = \hat{f}(Y)$ to satisfy $P(X|Y) = P(X|X')$. We then have $P(X, Y) = P(Y)P(X|X')$. Hence,

$$P(X|X') = \frac{P(X)P(Y|X)}{P(Y)}. \tag{4}$$

Because $|X'| < |Y|$, there must exist two values $y_1 \neq y_2$ such that $\hat{f}(y_1) = \hat{f}(y_2)$, which implies $P(X|\hat{f}(y_1)) = P(X|\hat{f}(y_2))$. According to Equation (4), we have

$$\frac{P(X)P(Y = y_1|X)}{P(Y = y_1)} = \frac{P(X)P(Y = y_2|X)}{P(Y = y_2)},$$

or

$$P(Y = y_1|X) = P(Y = y_2|X) \cdot \frac{P(Y = y_1)}{P(Y = y_2)},$$

which contradicts Assumption A1. Therefore, the reverse direction does not admit a low-cardinality hidden representation. $\square$

Note that assumption A1 may be violated, but the chance for it to be violated should be low. Roughly speaking, this assumption states that $X$ and $Y$ are not "locally" independent. Suppose assumption A1 does not hold; then there must exist $y_1, y_2$ satisfying $P(Y = y_1|X) = cP(Y = y_2|X)$ for all possible values of X. One can derive that $P(X|Y = y_1) = P(X|Y = y_2)$. This means if we ignore all the other possible values of $Y$ other than $y_1$ and $y_2$, $X$ and $Y$ become independent. Generally speaking, this will not hold when $X$ and $Y$ are dependent, especially when the cardinality of $X$ is not small. The experimental results also illustrate the plausibility of this assumption.

As an immediate result of Theorem 1, we have the identifiability of the causal direction under the hidden compact representation model, as given in Theorem 2.

**Theorem 2.** *Assume that in the causal direction there exists the transformation $Y' = f(X)$ such that $P(Y|X) = P(Y|Y')$, where $|Y'| < |X|$, and and assumption A1 holds. Then to produce the same distribution $P(X, Y)$, the reverse direction must involve more effective number of parameters in the model than the causal direction.*

Going one step further, Theorem 3 shows the BIC of the causal direction is asymptotically higher than that of the reverse one. The proof of this theorem is provided in the supplementary material.

**Theorem 3.** *If the reverse direction involves more parameters than the causal direction to produce the same distribution $P(X, Y)$, the BIC of the causal direction is asymptotically higher than that of the reverse one.*

# 4 Experiments

To investigate the effectiveness of the proposed method based on the hidden compact representation model, we compare it with baseline algorithms on both synthetic data and the real world data. On synthetic data, we simulate the data according to the hidden compact representation model. In all the experiments, we generate 1000 different causal pairs and 2000 samples for each pair. On real-world data, we run the algorithm on Pittsburgh Bridges dataset and Abalone dataset. The implementation of HCR can be found on CRAN [1].

The following five algorithms are taken as the baseline: ANM [Peters *et al.*, 2010], SA [Liu and Chan, 2016a], DC [Liu and Chan, 2016b], IGCI [Janzing *et al.*, 2012] and CISC [Budhathoki and Vreeken, 2017]. The parameter settings of all the algorithms are based on their origin work.

To make a fair comparison, the decision rate is used as the metric to evaluate the models' performance, same as that in IGCI [Janzing *et al.*, 2012] and CISC [Budhathoki and Vreeken, 2017].

## 4.1 Synthetic Data with Hidden Compact Representation Model

In this set of experiments, the samples are generated according to the following two-stage procedure. Firstly, generate $X$ from a multinomial distribution and its cardinality is randomly chosen from $\{3, 4, ..., 15\}$. Secondly, map each $X$ to a value that uniformly samples from the interval $\{1, 2, ..., |X|\}$. Finally, randomly generate a conditional probability distribution $P(Y|Y')$ and sample Y according to $Y'$ and $P(Y|Y')$, and $|Y|$ is generated from the interval $\{|Y'|, ..., 15\}$.

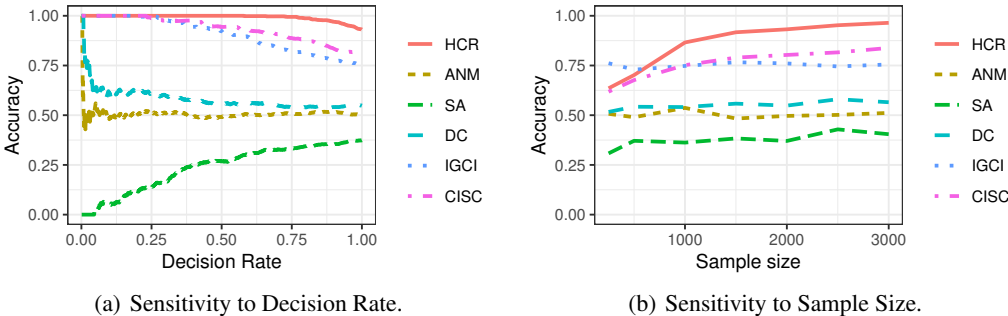

(a) Sensitivity to Decision Rate.
(b) Sensitivity to Sample Size.

Figure 2: Results on the Hidden Compact Representation Model.

Figure 2(a) shows the accuracy with difference decision rate. As shown in the figure 2(a), HCR outperforms the baseline methods across all the decision rates. HCR achieves acceptable results even when the decision rate is 1, which shows HCR can reliably infer the causal direction for all the cause-effect pairs. In this set of experiment, ANM fails to work because its additive noise assumption may not hold for the current causal mechanism.

Figure 2(b) shows the performance of the algorithms with the sample size varying from 250 to 3000. The decision rate is 1 in this set of experiments. As shown in the figure, the performance of HCR grows much faster than the baseline methods and converge to 1 when the sample size reaches 3000. This shows that the hidden compact representation explores the information behind the data more efficiently, compared with the other algorithms.

## 4.2 Real-World Data

To further assess the performance of our method for the discrete casual inference, we test the algorithms on two real-world datasets, *Pittsburgh Bridges dataset* and *Abalone dataset*. Both of them are wildly used in previous research and can be downloaded from UCI Machine Learning Repository [Lichman, 2013].

**Pittsburgh Bridges dataset**: There are 108 bridges in this dataset. The following 4 cause-effect pairs are known as ground truth in this experiment. They are 1) Erected (Crafts, Emerging, Mature, Modern) → Span (Long, Medium, Short), 2) Material (Steel, Iron, Wood) → Span (Long, Medium, Short); 3) Material (Steel, Iron, Wood) → Lanes (1, 2, 4, 6); 4) Purpose (Walk, Aqueduct, RR, Highway) → type (Wood, Suspen, Simple-T, Arch, Cantilev, CONT-T).

Table 1: Hidden Compact Representation on Pittsburgh Bridges Data Set.

| Ground truth | $f(X) \rightarrow Y'$ | $P(Y\|Y')$ |
|---|---|---|
| Erected→Span | $f(\{Crafts\}) \rightarrow 1$ | Medium: 0.5, Short:0.5 |
|  | $f(\{Emerging, Mature, Modern\}) \rightarrow 2$ | Long: 0.37, Medium:0.59, Short:0.04 |
| Material→Span | $f(\{Steel\}) \rightarrow 1$ | Long: 0.42, Medium:0.58 |
|  | $f(\{Iron, Wood\}) \rightarrow 2$ | Medium:0.55, Short:0.45 |
| Material→Lanes | $f(\{Steel\}) \rightarrow 1$ | 2 Lane:0.6, 4 Lane:0.33, 6 Lane:0.06 |
|  | $f(\{Iron, Wood\}) \rightarrow 2$ | 1 Lane:0.15, 2 Lane:0.8, 4 Lane:0.04 |
| Purpose→Type | $f(\{Aqueduct, Highway, Walk\}) \rightarrow 1$ | Arch:0.18, Cantilev:0.12, CONT-T:0.12, Simple-T:0.24, Suspen: 0.15, Wood:0.19 |
|  | $f(\{RR\}) \rightarrow 2$ | Cantilev:0.06, CONT-T:0.03, Simple-T:0.81, NIL:0.3,wood:0.06 |

Generally speaking, HCR can identify all 4 cause-effect pairs correctly. To gain an insight into the hidden compact representation, we give the reconstructed model in Table 1. In detail, the result on "Erected → Span" shows that {Emerging, Mature, and Modern} of erected are mapped into a hidden compact representation $Y' = 2$, while Crafts is mapped to $Y' = 1$. This hidden representation reflects that Crafts is the main cause of the medium and short bridge, which is compatible with common sense. Similarly, from the results on "Material → Span" and "Material → Lanes", we can see that the steel belongs to modern material with high strength, while iron and wood are classic materials with lower strength. This hidden property of the material causes the different span and lanes. Similar results can be found in "purpose → type". All of those results on the four cause-effect pairs reflect that HCR is a proper representation of the causal mechanism for discrete data.

Figure 3(a) shows the results of the algorithms on Pittsburgh Bridges dataset with different sample sizes. Because of the space limitation, only the average result of the four pairs are reported. As shown in the figure, HCR outperforms the baseline methods and shows competitive performance even when the sample size is smaller than 100, while the other four baselines are all failed to find the right causal direction with such a small sample size. This reflects HCR might be a suitable representation of the causal mechanism in this real-world scenario.

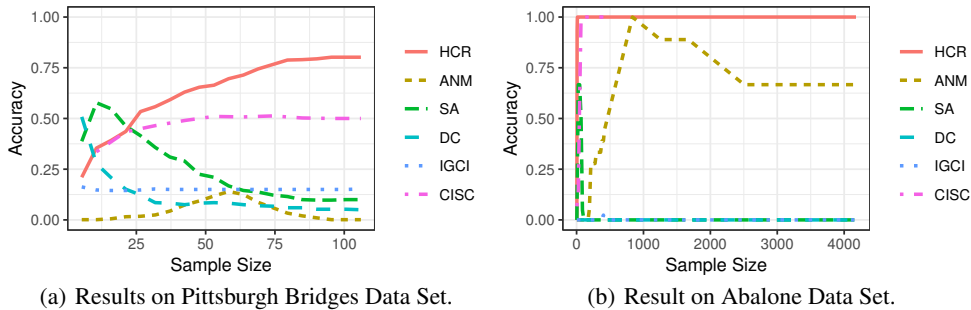

(a) Results on Pittsburgh Bridges Data Set.  (b) Result on Abalone Data Set.

Figure 3: Sensitivity to Sample Size.

**Abalone Data Set**: This dataset contains 4177 samples and each sample has 4 different properties. The ground truth contains three cause-effect pairs, Sex → {Length, Diameter, Height}. The property sex has three values, male, female and infant. The length, diameter, and height are measured in mm and treated as discrete values, similar to [Peters *et al.*, 2010].

Table 2: Hidden Compact Representation on Abalone Data Set.

| Ground truth | $f(X) \to Y'$ | $P(Y\|Y')$ |
|---|---|---|
| Sex→Length | $f(\{Infant\}) \to 1$ | $0.43 \pm 0.1$ |
| | $f(\{Female, Male\}) \to 2$ | $0.57 \pm 0.96$ |
| Sex→Diameter | $f(\{Infant\}) \to 1$ | $0.33 \pm 0.088$ |
| | $f(\{Female, Male\}) \to 2$ | $0.45 \pm 0.079$ |
| Sex→Height | $f(\{Infant\}) \to 1$ | $0.11 \pm 0.032$ |
| | $f(\{Female, Male\}) \to 2$ | $0.15 \pm 0.037$ |

In this dataset, HCR successfully determines the causal direction for all the three pairs, and the details of the model are given in Table 2. Because the properties have many discrete states, column $Y$ shows its mean and standard variance. Although this dataset closely relates to the additive noise model, Table 2 demonstrates that HCR still successfully identifies the causal direction and provides a fruitful insight of the causal mechanism. In detail, the model maps Infant to $Y' = 1$, and maps {Female, Male} to $Y' = 2$. Here $Y'$ indicates that categorizing the sex of abalones into male and female is redundant relative to the considered effect, which is Length, Diameter, or Height. In the second stage, the mapping shows the maturity causes the sizes.

We also compare our results with the baseline methods with different sample sizes. As shown in Figure 3(b), although this dataset follows the assumptions of the discrete additive noise model, HCR still outperforms ANM. Note that, ICGI and CISC achieve the same performance as HCR and their curves are covered by that of HCR. Moreover, SA and DC fail to give the correct direction on this data set, perhaps because they are designed for the discrete data with a small number of cardinalities while the cardinality of the variables is very large in this dataset. These results also indicate that HCR may provide a suitable representation of the causal mechanism in various scenarios.

As a summary, HCR stably outperforms all the baseline methods on these two real-world discrete datasets and, furthermore, shows a meaningful hidden compact representation of the causal mechanism.

## 5 Conclusion

Finding causal direction between discrete variables is an important but challenging problem. In this paper, we make an attempt to solve this problem by developing a low-cardinality hidden representation model for the causal mechanism, which decomposes the mechanism into two stages. With this model estimated by the Bayesian Information Criterion (BIC), we develop an effective causal discovery method for discrete variables. Theoretical analysis also shows that the model is generally identifiable—it is not identifiable when some weak technical conditions on the causal mechanism are violated. Experimental results on both synthetic and real data verify our theoretical results and support the validity of the proposed model, at least in a number of real situations. In future work, we plan to extend the proposed method to discrete data with confounding factors.

## Acknowledgments

This research was supported in part by NSFC-Guangdong Joint Found (U1501254), Natural Science Foundation of China (61876043, 61472089), NSF of Guangdong (2014A030306004, 2014A030308008), Science and Technology Planning Project of Guangdong (2015B010108006, 2015B010131015), Guangdong High-level Personnel of Special Support Program (2015TQ01X140) , Pearl River S&T Nova Program of Guangzhou (201610010101). This material is partially based upon work supported by United States Air Force under Contract No. FA8650-17-C-7715, by National Science Foundation under EAGER Grant No. IIS-1829681, and National Institutes of Health under Contract No. NIH-1R01EB022858-01, FAINR01EB022858, NIH-1R01LM012087, NIH-5U54HG008540-02, and FAIN-U54HG008540, and work funded and supported by the Department of Defense under Contract No. FA8702-15-D-0002 with Carnegie

Mellon University for the operation of the Software Engineering Institute, a federally funded research and development center. Any opinions, findings, and conclusions or recommendations expressed in this material are those of the authors and do not necessarily reflect the views of the United States Air Force or the National Institutes of Health or the National Science Foundation. We appreciate the comments from anonymous reviewers, which greatly helped to improve the paper.

## Footnotes

[1] https://cran.r-project.org/package=HCR

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
