[Supplementary Material]

# Supplementary Material for Causal Discovery from Discrete Data using Hidden Compact Representation

**Ruichu Cai [1], Jie Qiao[1], Kun Zhang[2], Zhenjie Zhang[3], Zhifeng Hao[1, 4]**

[1] School of Computer Science, Guangdong University of Technology, China
[2] Department of philosophy, Carnegie Mellon University
[3] Singapore R&D, Yitu Technology Ltd.
[4] School of Mathematics and Big Data, Foshan University, China
cairuichu@gdut.edu.cn, qiaojie.chn@gmail.com, kunz1@andrew.cmu.edu,
zhenjie.zhang@yitu-inc.com, zfhao@gdut.edu.cn

**Theorem 2.** *Assume that in the causal direction there exists the transformation $Y' = f(X)$ such that $P(Y|X) = P(Y|Y')$, where $|Y'| < |X|$, and and assumption A1 holds. Then to produce the same distribution $P(X, Y)$, the reverse direction must involve more effective number of parameters in the model than the causal direction.*

*Proof.* Recall that the parameter for the causal direction is $d_{X \to Y} = |X| - 1 + |Y'|(|Y| - 1)$. Since $|Y'| < |X|$, we conclude $d_{X \to Y} < |X| - 1 + |X|(|Y| - 1)$. For reverse direction, the number of effective parameters is $d_{Y \to X} = |Y| - 1 + |X'|(|X| - 1)$. Based on Theorem 1, the reverse direction does not admit a low-cardinality hidden representation. Hence, we have $|X'| = |Y|$. Compare the difference between $d_{X \to Y}$ and $d_{Y \to X}$, we have:

$$d_{X \to Y} - d_{Y \to X} < |X| - 1 + |X|(|Y| - 1) - (|Y| - 1 + |Y|(|X| - 1))$$
$$= 0 \tag{1}$$

Thus, reverse direction must involve more effective number of parameters than the causal direction. $\square$

**Theorem 3.** *If the reverse direction involves more parameters than the causal direction to produce the same distribution $P(X, Y)$, the BIC of the causal direction is asymptotically higher than that of the reverse one.*

*Proof.* First, we will proof that the likelihood on the causal direction asymptotically greater or equal than that of the reverse direction. Equation (2) gives the gap between the causal direction $M$ and the reverse direction $\hat{M}$.

$$
\begin{aligned}
&\mathcal{L}(\mathcal{D}; M) - \mathcal{L}(\mathcal{D}; \hat{M}) \\
&= \sum_{i=1}^{m} \log \frac{P(X = x_i) P(Y = y_i | Y' = f(x_i))}{P(Y = y_i) P\left(X = x_i | X' = \hat{f}(y_i)\right)} \\
&= \sum_{i=1}^{m} \log \frac{P(X = x_i, Y = y_i)}{P(Y = y_i) P\left(X = x_i | X' = \hat{f}(y_i)\right)} \\
&= m E_{x, y \sim p(x, y)} \left( \log \frac{P(X = x, Y = y)}{P(Y = y) P\left(X = x | X' = \hat{f}(y)\right)} \right) \\
&= m KL\left( P(X = x, Y = y) || P(Y = y) P\left(X = x | X' = \hat{f}(y)\right) \right) \\
&\geq 0
\end{aligned}
\tag{2}
$$

The first equality is based on the likelihood of the HCR model. The second equality is based on the fact that $X \perp\!\!\!\perp Y | Y'$. The third equality is based on the existing of sufficient number of samples. The forth equality and the fifth inequality are based on the definition and the property of KL-divergence respectively.

Second, we will show that the penalty on the causal direction less than the reverse direction. Because the causal direction $M$ and the reverse direction $\hat{M}$ produce the same probability $P(X, Y)$, and the inverse direction $\hat{M}$ involves more parameters than the causal direction $M$ according to the Theorem 2.

Thus, the causal direction's BIC is asymptotically higher than the reverse ones. $\qquad\square$