[Reviews · NeurIPS 2018]

Reviewer 1



UPDATE after authors' rebuttal: Thanks a lot for your detailed answers, quite convinced by most of your arguments, you played the game well within the one page constraint! I increase my score from 6 to 7, hoping it would be enough to see your paper being presented at NIPS! If not (competition is strong here!), I strongly encourage you to still fight to publish it, clearly a key topic (I'm biased :)). This work presents a method to decipher a causal link between 2 variables. The method relies on a re-coding of one of the variables (the cause) to obtain an 'optimal' link with the effect variable. The paper reads well, is well structured and easy to understand. I am puzzled as to whether it is a mere simple implementation of existing work or whether it is a truly novel (and then would well be worth presenting) method. The numerical results are not very convincing. Comments: - abstract: as a first pass, I was intrigued by the use of a hidden variable mentioned here, wondered whether it was unobserved. OK once reading the paper. - introduction: l36 "right" -> define what the right order is. Is the variable necessarily ordered? Cannot be purely qualitative? This paragraph is quite interesting. Do you have a ref? If this is original work, may need further explanation. l45 (typo): ...maps THE cause to...of lower cardinality - Before you introduce your work from l43, is the motivation strong enough? I am not entirely convinced at this stage...Moreover, it reads that a (simple) re-coding of the (true) cause is enough to distinguish the true causal relationship. It is not very intuitive. Careful, the Table of Figure 2 is not causal, only probabilistic (see text l60) - Section 2, l72: f needs to be a surjection, right? You could enumerate them (how many are there for each unknow |Y^{\prime}| ? l79: not sure your example is a misdiagnosis example. l83: LOG-likelihood. l96: ...$ P(X) $, THE deterministic... - l103: is the alternate maximisation equivalent to a joint maximisation or an approximation? Quality of the approximation then? l117: ...with $ y_0 $, where...Could you also explain the choice for y0? - Bottom of p4: so in the end, (i) re-coding either X or Y and finding a 'good' probabilistic relationship with the target variable is enough? (ii) Moreover, it quantifies the 'correctness' of the model? (iii) What if neither X->Y, nor Y->X are correct? - Section 3: does Th 1 generalise the additive noise model assumption? THIS IS INTERESTING. So is Th 4. - Section 4: l174, remove (HCR in short), l175 ...on both synthetic and real world data. Rephrase the next sentence. - Your choice is synthetic data looks 'easy' as it is exactly your model. May need more general framework! You could then do some sensitivity of your method to breaking the assumption of Th 1?! - l188: notation issue, do you mean {3,...,15} instead? Fig2: why don't you show performance below m=500? Strange performance for SA, ANM and DC. Worse than random guess??!! Define the decision rate. - real world data: I am sure these data sets were used in some public challenges (see the work by I. Guyon), I doubt they get results as poor as the ones you show. How do you explain some method perform more poorly when m increases? - References: consistency, with or without first names? (e.g. Jonas Peters, but B. Sch\"{o}lkopf)

Reviewer 2



In this manuscript, the authors propose to infer causal relations from categorical observational data by assuming a two-stage causal discover process and finding a hidden compact representation of the cause. The manuscript also provides an effective solution to recover the compact representation. The effectiveness of the proposed approach is verified in both synthetic and real-world data. Overall, I highly appreciate the manuscript because of following strengths: 1. Causal discovery on categorical data is an important yet challenging problem to be addressed in the literature. 2. The approach is well motivated with an illustrating example. The writing and logic is clear. 3. The manuscript also shows that using the model, the causal direction is identifiable in the general case. 4. The experimental results are promising. I have only several minor comments: 1. The manuscript should make it clear what the space of Y^{\prime} is. Is it also a categorial variable? I can not tell this when reading the first three pages. 2. It’s better to use some examples to help illustrate the identifiability part. 3. A brief introduction to the benchmarks is needed. What kind of data setting are they focusing on?

Reviewer 3



The authors propose a procedure for identifying causal relationships between bivariate categorical data. In particular, they assume that the causal relationship can be explained by an intermediate lower dimensional representation. Major Comments: 1) For the HCR to be identifiable, must |Y'| < |X|? If |Y'| = |X| = |Y| it seems that the model would be invertible, but the assumption that |Y'| must be strictly less than |X| is never given. 2) Perhaps I am missing something, but is Theorem 2 a direct implication of Theorem 1? There does not seem to be a proof provided. 3) For the Real-world data sets, it would be interesting to see if an exhaustive search over the |X|! partitions for f results in a different result than the greedy search. Minor comments: Line 174: Strikethrough Line 177: "... and are 2000 samples for each pair" presumably refers to Figure 2(a) only as Figure 2(b) shows varying sample sizes UPDATE: Thanks for the responses. Would be nice to see if the greedy vs exhaustive search differs in simulation study (not simply on a single data set), but not necessary and more of a "nice-to-have." Interesting work and would like to see this at NIPS